# Photothermal Regulated Nanozyme of CuFeS_2_ Nanoparticles for Efficiently Promoting Wound Healing Infected by Multidrug Resistant Bacteria

**DOI:** 10.3390/nano12142469

**Published:** 2022-07-19

**Authors:** Zezhong Liu, Zengxu Liu, Zhen Zhao, Danxia Li, Pengfei Zhang, Yanfang Zhang, Xiangyong Liu, Xiaoteng Ding, Yuanhong Xu

**Affiliations:** Institute of Biomedical Engineering, College of Life Sciences, Qingdao University, 308 Ningxia Road, Qingdao 266071, China; lzz56066@163.com (Z.L.); zengxu1996l@163.com (Z.L.); zhaozhenciac@163.com (Z.Z.); ldxjy2018@163.com (D.L.); pfzhang0106@163.com (P.Z.); zyf183162@163.com (Y.Z.); z337047@163.com (X.L.); yhxu@qdu.edu.cn (Y.X.)

**Keywords:** bacterial infection, CuFeS_2_ NPs, peroxidase, photothermal effect, combined antibacterial

## Abstract

Peroxidase-mediated chemokinetic therapy (CDT) can effectively resist bacteria; however, factors such as the high dosage of drugs seriously limit the antibacterial effect. Herein, CuFeS_2_ nanoparticles (NPs) nanozyme antibacterial system with the photothermal effect and peroxidase-like catalytic activity are proposed as a combined antibacterial agent with biosafety, high-efficiency, and broad-spectrum antibacterial ability. In addition, the as-obtained CuFeS_2_ NPs with a low doses of Cu^+^ and Fe^3+^ can change the permeability of bacterial cell membranes and break the antioxidant balance by consuming intracellular glutathione (GSH), which results in more conducive ROS production. Meanwhile, the photothermal heating can regulate the CuFeS_2_ NPs close to their optimal reaction temperature (60 °C) to release more hydroxyl radical in low concentrations of H_2_O_2_ (100 µM). The proposed CuFeS_2_ NPs-based antibacterial system achieve more than 99% inactivation efficiency of methicillin-resistant Staphylococcus aureus (10^6^ CFU mL^−1^ MRSA), hyperspectral bacteria β-Escherichia coli (10^6^ CFU mL^−1^ ESBL) and Pseudomonas aeruginosa (10^6^ CFU mL^−1^ PA), even at low concentration (2 μg mL^−1^), which is superior to those of the conventional CuO NPs at 4 mg mL^−1^ reported in the literature. In vivo experiments further confirm that CuFeS_2_ NPs can effectively treat wounds infected by MRSA and promote the wound healing. This study demonstrates that excellent antibacterial ability and good biocompatibility make CuFeS_2_ NPs a potential anti-infection nanozyme with broad application prospects.

## 1. Introduction

With the continuous development of drug-resistant strains, bacterial infectious diseases are a major threat to global health [1,2,3]. In traditional treatment strategy, antibiotics are the most effective and conventional method for the treatment of bacterial infection. However, the heavy use of antibiotics has led to the occurrence of multiple drug resistance bacteria, which causes a sharp decline in the efficacy of antibiotics [4]. Consequently, effective antimicrobial agents are severely needed, as a substitute to antibiotics. The development of new antibacterial platforms based on nanomaterials, Ag- or Cu-based, have been well-developed and regarded as efficiently antibacterial, but high-dose applications still suffered from potentially non-negligible biological toxicity, due to their intrinsic properties of heavy metals. Owing to the advantages of high catalytic activity, substrate selectivity, higher stability, and adaptability to special environments [5], nanozymes have been diffusely applied in the areas of antibacterial treatment, in order to treat the infected wounds in animals [6,7,8]. For example, peroxidase mimics can specifically facilitate the conversion of hydrogen peroxide (H_2_O_2_) to the more oxidizing hydroxyl radical (•OH), in order to achieve a better antibacterial effect, thus effectively avoiding the side effects of high concentration of H_2_O_2_ to animal healthy tissues [9,10]. Some inorganic nanomaterials with peroxidase-like activity have been reported, such as V_2_O_5_ QDs, Au NPs, and MoS_2_ nanomaterials, etc. [3,11]. Although these nanomaterials have shown great potential for antimicrobial use in peroxidase-mediated chemokinetic therapy (CDT), their antibacterial effects are limited, due to the insufficient production of hydroxyl radicals (•OH) and short effective action distance of hydroxyl radicals [12]. Photothermal therapy (PTT), which can use local near-infrared (NIR) light to cause high temperatures or improve the catalytic activity of nanozymes, which has been diffusely used in mice tumor treatment and antibacterial because of their advantages, i.e., less invasiveness, fewer side effects, and better controllability, compared with other antibacterial drugs. However, long time exposure to high power density NIR may induce skin destruction and bacterial heat resistance [13]. Since the antibacterial therapy alone is not efficient enough to reach satisfying efficiency, a combined therapy strategy is expected to compensate for each other, in order to overcome the shortcomings of single-mode therapy, thus greatly improving the antibacterial effect [14]. While enhancing the antibacterial efficiency of CDT, by means of multi-mode combination and ROS release, the antioxidant system of bacterial cells should not be ignored [15]. Glutathione (GSH) is an intracellular reducing substance that consumes hydroxyl radicals to inhibit the therapeutic effect of CDT. Therefore, the consumption of GSH helps to further improve the therapeutic effect of CDT [16,17]. In order to treat bacterial-infected wounds and promote wound healing, it is necessary to develop efficient and safe nano antibacterial materials. In addition, the efficient nanozyme against pathogenic bacteria with the synergistic effect of CDT, PTT, and GSH consumption capacity has rarely been reported.

Herein, we prepared and characterized the CuFeS_2_ nanoparticles (NPs) and evaluated the antibacterial efficacy against both Gram-positive and -negative bacteria. Under the irradiation of a NIR laser, the activity of CuFeS_2_ NPs peroxidase is effectively enhanced, under the condition that CuFeS_2_ does not produce temperatures that are too high, and more hydroxyl radicals are released, so as to kill pathogenic bacteria more effectively. In addition, under simulated physiological conditions, CuFeS_2_ NPs showed the ability to consume GSH. Based on the above advantages, the as-obtained CuFeS_2_ NPs can consume the GSH in bacteria and effectively fight pathogenic bacteria via the combination of PTT- and ROS-mediated CDT (Figure 1). High-resolution transmission electron microscope (HRTEM) and transmission electron microscope (TEM) were used to characterize the morphology. The UV–VIS absorption spectrum and fluorescence methods were used to detect the hydroxyl radical and characterize the enzyme activity of CuFeS_2_ NPs.

## 2. Materials and Methods

### 2.1. Synthesis of CuFeS_2_ Nanoparticles

With deionized water as the solvent, 50 mL of deionized water was taken into a conical flask. The water bath was heated to 90 °C for more than 5 min, in order to effectively eliminate dissolved oxygen. Next, FeSO_4_·7H_2_O (5.6 mg) and CuCl_2_·2H_2_O (17.0 mg) were dispersed into the deionized water (1.2 mL), adjusting the pH of the mixed solution to about 12.0. Then, Na_2_S·9H_2_O (97.0 mg) was quickly added into the above mixed solution directly, and the solution turned black. Finally, the samples were converged by centrifugation, washed three times with deionized water, and freeze-dried for use.

### 2.2. POD-like Activity and Kinetic Assay

In order to explore the peroxidase-mimic activity of CuFeS_2_ NPs, TMB was selected as the chromogenic substrate, and the UV absorption changes (652 nm) of the reaction system were observed and recorded in the presence of H_2_O_2_ in the acetate buffer (0.1 M, pH 4.0). Accordingly, 0.1 mL CuFeS_2_ NPs suspension (40 μg mL^−1^), 0.025 mL H_2_O_2_ (0.5 mM), and 0.025 mL TMB (0.5 mM) were added to 0.85 mL acetate buffer. A total of four groups of reaction systems were set: TMB + H_2_O_2_, CuFeS_2_ + TMB, CuFeS_2_ + H_2_O_2_, and TMB + CuFeS_2_ + H_2_O_2_ at room temperature for 5 min. Then, the color pictures of each reaction system were taken, and the UV spectra of different systems were scanned by ultraviolet spectrophotometer. The temperature (22~90 °C) and pH (2.0~11.0) tolerance of the peroxidase of CuFeS_2_ NPs were studied with an acetate buffer (pH 4.0, 0.1 M) as the background, and the maximum absorbance of each group was defined as 100%. The peroxidase kinetics of CuFeS_2_ NPs was studied by absorbance monitoring at 652 nm under different conditions. In the pH 4.0 acetate buffer, the concentration of CuFeS_2_ NPs in the reaction system was 40 μg mL^−1^, and the concentration of H_2_O_2_ or TMB of one substrate was fixed, respectively, while the concentration of the other substrate was changed. The total system was 1 mL, and the reaction time was 10 min. After 10 min of reaction, the absorbance of each system was measured. The kinetic parameters of CuFeS_2_ NPs peroxidase were calculated according to the Michaelis equation:(1)$$\frac{1}{V}=\frac{K_{m}}{V_{\max}}\times \frac{1}{\left[S\right]}+\frac{1}{V_{\max}}$$
where K_m_ delegates the Mie constant, [S] delegates the substrate concentration, and V_max_ stands for the maximum reaction rate. The reaction rate and substrate concentration were fitted by the Michaelis–Menten equation, V_max_ and the kinetic parameters Km were obtained by Lineweave–Burk plot.

### 2.3. Detection of Hydroxyl Radical

The hydroxyl radical was detected by fluorescence method. In order to detect •OH in the reaction catalyzed by CuFeS_2_ NPs peroxidase, TA was selected as a fluorescent probe. TA itself had no fluorescence. When TA captured •OH, it formed 2-hydroxyl terephthalic acid (TAOH) with a strong fluorescence emission peak at 435 nm. In phosphate buffer solution (pH 7.4, 0.1 M), the concentration of each component in the reaction system was CuFeS_2_ NPs (40 μg mL^−1^), Ta (0.5 mM), and H_2_O_2_ (1.0 mM). Four groups of reaction systems were set: Ta, Ta + H_2_O_2_, Ta + CuFeS_2_, and Ta + H_2_O_2_ + CuFeS_2_. The samples were placed at room temperature for 12 h, and the fluorescence spectra of each sample were measured by a fluorometer (Edinburgh FS5).

Electron spin resonance spectroscopy (ESR) measurements were made at room temperature on a standard X-band Bruker E-500 EPR spectrometer (Germany). DMPO (5, 5-dimethyl-1-pyrrolin-N-oxide) was used as a trap to capture free •OH to form DMPO/•OH spin adducts. The concentrations of each component in the system were H_2_O_2_ (10 mM), CuFeS_2_ (40 μg mL^−1^), and DMPO (40 mM). Four groups of samples were set to be tested: DMPO, DMPO + H_2_O_2_, DMPO + CuFeS_2_, and DMPO + H_2_O_2_ + CuFeS_2_. The prepared sample solution was transferred into a quartz capillary tube and placed for 30 min to monitor the ESR signal. All of the above operations were performed in a NAAC buffer (0.1 M, pH 4.0). All ESR spectra were obtained with 1 G field modulation, 100 G scanning range, and 20 mW microwave power.

### 2.4. Bacterial Culture and In Vitro Bacteriostatic Experiment

Bacterial culture and in vitro bacteriostatic experiment. Pseudomonas aeruginosa (PA), hyperspectral β-lactamase-producing Escherichia coli (ESBL), and methicillin-resistant Staphylococcus aureus (MRSA) were inoculated in LB liquid medium (37 °C, 180 rpm) for 12 h, respectively. The obtained bacterial solution was diluted to 10^6^ CFU mL^−1^, and the following treatments were performed: PBS, H_2_O_2_, CuFeS_2_, CuFeS_2_ + H_2_O_2_, PBS + NIR, H_2_O_2_ + NIR, CuFeS_2_ + NIR, and CuFeS_2_ + H_2_O_2_ + NIR, which were treated with NIR (1 W cm^−2^, 808 nm) for 5 min. The total system of each experimental group was 300 μL, including 10^6^ CFU mL^−1^ bacterial suspension, 2 μg mL^−1^ CuFeS_2_ NPs, and 0.1 mM H_2_O_2_. After incubation at 37 °C for 30 min, 100 μL of the treated bacterial solution was taken and evenly coated on the plate. The bacteria were incubated overnight at 37 °C, observed, and photographed.

### 2.5. Fluorescence Detection of Living/Dead Bacteria

In order to visually identify the live/dead state of the bacteria, the bacterial samples treated with different methods were fluorescently stained. The concentration of bacterial liquid was 10^8^ CFU mL^−1^, and 8 groups of experiments were set for the following treatments: PBS, H_2_O_2_, CuFeS_2_, CuFeS_2_ + H_2_O_2_, PBS + NIR, H_2_O_2_ + NIR, CuFeS_2_ + NIR, CuFeS_2_ + H_2_O_2_ + NIR, NIR refers to under the irradiation of 808 nm NIR light (808 nm, 1 W cm^−2^) for5 min and incubated with bacteria at 37 °C for 30 min. Then, the unbounded dye was washed using PBS after being incubated at 37 °C for 15 min using SYTO9 and PI fluorescent dye. Drop-coated glass slides were used for sample preparation. Bacteria were observed under a positive fluorescence microscope, and fluorescence staining images were taken. Bacteria in normal state showed green fluorescence, while bacteria in dead state showed red fluorescence.

### 2.6. Bacterial ROS Fluorescence Detection

DCFH-DA (10 μm) was added into 20 mL 0.9% NaCl solution and co-incubated with 10^8^ CFU/mL bacteria for 30 min. The combined DCFH-DA was removed by centrifugation at 8000 RPM for 5 min. The bacterial cells were treated in different ways, as follows: PBS, H_2_O_2_, CuFeS_2_, CuFeS_2_ + H_2_O_2_, PBS + NIR, H_2_O_2_ + NIR, CuFeS_2_ + NIR, and CuFeS_2_ + H_2_O_2_ + NIR were treated with Nir (808 nm, 1 W cm^−2^) for 5 min and incubated at 37 °C for 4 h. Fluorescence intensity was measured with a fluorometer and observed and photographed under a fluorescence microscope.

### 2.7. SEM Characterization of Bacteria

Methicillin-resistant Staphylococcus aureus (MRSA), hyperspectral β-lactamase-producing Escherichia coli (ESBL), and Pseudomonas aeruginosa (PA) were treated as follows: after incubation with CuFeS_2_ + H_2_O_2_ at 37 °C for 4 h, each reaction system was centrifuged at 3000 RPM for 5 min, and then washed with PBS for 3 times. Then, they were fixed with 2.5% glutaraldehyde, overnight at 4 °C, followed by dehydration with ethanol solution of different concentrations, followed by treatment with 50% and 100% isoamyl acetate, the final zero-point drying, smear sample preparation.

### 2.8. Biocompatibility Evaluation of CuFeS_2_ Nanomaterials

The biological toxicity of CuFeS_2_ nanomaterials was evaluated by hemolysis test. The specific procedures were as follows. Fresh blood of rats was taken, centrifuged (15 min, 1500 rpm) to concentrate red blood cells, and washed with normal saline for 3 times. The red blood cell suspension was diluted to 5%, treated with normal saline, water, and different concentrations of CuFeS_2_ NPs (25, 50, and 100 μg mL^−1^), incubated for 3 h at 37 °C, and centrifuged (5 min, 11,000 rpm). The absorbance value of the supernatant at 540 nm was determined. The hemolysis rate of CuFeS_2_ NPs was computed with water and normal saline treatment groups as controls.

MTT assay was carried out to assess the cytotoxicity of CuFeS_2_ NPs using L929 fibroblasts. The procedure was as follows. Approximately 5000 cells were inoculated in each well in a 96-well plate medium containing 10% FBS and 1% penicillin-streptomycin. The CO_2_ cell incubator was cultured until the cells were filled, and different concentrations of CuFeS_2_ NPs samples were added to continue the culture. After incubation for 24 h, 20 μL 5 mg mL^−1^ MTT were added to each well and continued to incubate for 4 h. Then, the obtained purple formazan was dissolved by DMSO (150 µL per well) with gentle shaking for 10 min. The absorbance at 490 nm was determined, and the cell viability was calculated.

In order to evaluate the compatibility of CuFeS_2_ in vivo, samples with concentrations of 40 and 80 μg mL^−1^ were prepared and injected into rats by intraperitoneal injection. The control group was given with the same volume of PBS buffer. After one-week treatment, the treated rats were killed, and the liver, heart, spleen, kidney, lung, and other organs were removed and fixed with formaldehyde; paraffin sections were made for histological analysis. At the same time, serum samples were collected to determine physiological indexes.

### 2.9. The Rats Woud Model

The Sprague–Dawley (SD) rats were aged 6 weeks, with a mean weight of 180 g. All the rats were randomly distributed into eight groups (each group of three) and slashed with about 0.6 cm wound area on their backs. After that, the wound area of the rats was injected with 1 × 10^6^ of *S. aureus*, which was then treated with different antibacterial groups of PBS, H_2_O_2_, CuFeS_2_, CuFeS_2_ + H_2_O_2_, PBS + NIR, H_2_O_2_ + NIR, CuFeS_2_ + NIR, and CuFeS_2_ + H_2_O_2_ + NIR. To harvest the wound tissue, the mice were sacrificed on the seventh day of the experiment, and the skin was excised, including the entire wound with adjacent normal skin. For histological examination, the wound tissues were gathered after the seventh day of the experiment and fixed in 4% paraformaldehyde solution. Then, the tissue samples were embedded in paraffin and stained with hematoxylin and eosin (H&E).

## 3. Results and Discussion

### 3.1. Preparation and Characterization of CuFeS_2_ NPs

The antibacterial nanomaterials were synthesized using CuCl_2_·2H_2_O, FeSO_4_·7H_2_O, and Na_2_S·9H_2_O as the precursors [18]. The morphology of the as-prepared CuFeS_2_ NPs was characterized by TEM (Figure 1a) and HRTEM (Figure 1b). It can be seen that the as-obtained nanoparticles have uniform size, and the particle size is calculated to be 10.38 ± 1.39 nm (Appendix A), as computed to counting the CuFeS_2_ particles of the TEM images. Meanwhile, the HRTEM image shows that CuFeS_2_ NPs have a parallel and ordered crystal lattice with 0.305 nm lattice spacing, corresponding to the (111) crystal of CuFeS_2_ (JCPDS 41-1404), which fits in with the literature reports [18]. In order to demonstrate the crystal characteristics of the product, XRD was performed, and the diffraction peaks were basically consistent with the standard card of the CuFeS_2_ crystal and in good agreement with the HRTEM data results (Figure 1c). The above results preliminarily prove that the CuFeS_2_ NPs was successfully synthesized. In order to study the element composition and valence of CuFeS_2_ NPs, XPS analysis was performed. According to the XPS spectra (Figure 1d), the samples were composed of Cu, Fe, and S elements, and their corresponding signal peaks were located at 931.81 eV, 710.70 eV, and 161.90 eV, respectively [19]. The high-resolution XPS spectra of Fe, S, and Cu for CuFeS_2_ NPs were analysed, respectively. Figure 1e shows the high-resolution spectrogram of Cu 2p; the peaks of 2p^1/2^ and Cu_2p_^3/2^ at 952.70 eV and 932.90 eV were obtained by peak separation fitting, which are attributed to Cu(I) in CuFeS_2_ [20]. Anyway, there was no signal peak at 942 eV, reflecting that there was almost no Cu(II) in the prepared CuFeS_2_, which is consistent with the previously reported results that only monovalent copper atoms exist in chalcopyrite structures [21]. Figure 1f is the high-resolution XPS spectrum of Fe 2p, the peaks at 724.83 eV and 711.15 eV are attributed to Fe^3+^_2p_^1/2^ and Fe^3+^_2p_^3/2^, while the double peaks at 713.49 eV and 719.49 eV are attributed to Fe(II) [22,23,24]. Figure 1g shows the high-resolution spectrum of S 2p, and the peak at 162.18 eV is attributed to the presence of S^2-^. The peak at 163.97 eV belongs to the polysulfide (S_n_^2-^) formed by sulphur vacancy on the surface of CuFeS_2._ The peaks at 168.66 eV and 169.74 eV are attributed to the presence of sulfate, possibly due to the oxidation of on the CuFeS_2_ surfaces [25,26]. The XPS characterization further confirmed that CuFeS_2_ was successfully synthesized, which is consistent with the XRD and HRTEM results. The photothermal conversion ability of the CuFeS_2_ NPs was studied by irradiating CuFeS_2_ NPs aqueous solution with an 808 nm NIR laser at 1.0 W cm^−2^. Figure 1h shows that the CuFeS_2_ NPs have effective photo thermal heating, and the corresponding solution temperature increased with the concentration of CuFeS_2_ NPs. In the presence of CuFeS_2_ NPs (1000 μg mL^−1^) and under 60 s NIR laser irradiation, the dispersion temperature increased to 50 °C (close to their optimum enzymatic temperature), demonstrating that CuFeS_2_ NPs have excellent photothermal conversion ability. In Appendix A, the CuFeS_2_ NPs still maintain good photothermal conversion ability after five heating/cooling processes under NIR laser irradiation, which proves the excellent photothermal stability of CuFeS_2_ NPs. According to Appendix A, the photothermal conversion efficiency [27] is further computed to be 29.8%, which is higher than that of traditional Au NPs (22.1%) [28]. The near infrared absorption of material with different concentrations also illustrated in Appendix A. These results proved that the prepared CuFeS_2_ NPs could be used as a potential PTT material.

### 3.2. CuFeS_2_ NPs Peroxidase Activity and Hydroxyl Radical Production

To study the peroxidase-like catalytic properties of CuFeS_2_ NPs, the ultraviolet spectra were used to measure the absorbance of TMB, ABTS, and OPD as color substrates in the existence of H_2_O_2_. The three substrates all produced specific color changes, and their corresponding characteristic absorption peaks were also observed in Figure 2a, which preliminarily proved that CuFeS_2_ NPs had intrinsic peroxidase activity. The prepared material can facilitate the reaction of H_2_O_2_ to produce hydroxyl radical, leading to the oxidation of TMB to ox-TMB, which changes the dispersion from colorless to blue. Compared with the control group, with only TMB or H_2_O_2_, the reaction solution could change from colorless to dark blue, producing a strong UV absorption peak at 652 nm upon the addition of both TMB and H_2_O_2_ (Figure 2b). In addition, the absorbance of the reaction system at 652 nm increased monotonically with the concentration of CuFeS_2_ NPs (Figure 2f), and the UV–VIS absorption spectra of reaction systems containing different concentrations of CuFeS_2_ (0, 6.25, 12.5, 25.0, 50.0, and 100 μg mL^−1^), indicating that the conversion of H_2_O_2_ to hydroxyl radical (•OH) was accelerated, which led to the acceleration of TMB oxidation. To further verify the peroxidase catalytic activity of CuFeS_2_ NPs, the hydroxyl radical generated in the catalytic process was monitored via the terephthalic acid (TA)-based fluorescent assays. As shown in Figure 2c and Appendix A, a fluorescence peak appears at 435 nm in the presence of CuFeS_2_ NPs, which reflected the generation of hydroxyl radicals. After the addition of H_2_O_2_, the fluorescence intensity improved much more apparently, reflecting that CuFeS_2_ NPs could convert H_2_O_2_ to a hydroxyl radical. The hydroxyl radical produced in the catalysis of CuFeS_2_ peroxidase was further determined by ESR characterization. As can be seen from Figure 2d, four typical signal peaks, with an intensity ratio of 1:2:2:1, can be obtained when CuFeS_2_ and H_2_O_2_ are present in the DMPO system, significantly different from CuFeS_2_ or H_2_O_2_ alone, which proves the production of the hydroxyl radical in the system. All the results confirmed the POD-like activity of the as-obtained CuFeS_2_ NPs. Additionally, we verified the influence of NIR on the peroxidase catalytic activity of CuFeS_2_ NPs. As shown in Figure 2e and Appendix A, the absorbance of the reaction system CuFeS_2_ + H_2_O_2_ + TMB was significantly increased under NIR illumination, and a higher fluorescence intensity was detected by the TA method, which indicates that the peroxidase-like activity of CuFeS_2_ NPs can be further enhanced upon the NIR light irradiation.

### 3.3. Kinetic Studies of the Enzyme-Mimic Activities of the As-Prepared CuFeS_2_

The impacts of pH and temperature on the peroxidase-like activity of CuFeS_2_ NPs were explored. As shown in Figure 3a,b, the catalytic activity was good in the pH range (2.0–11.0) and a wide temperature range (22–90 °C), and the optimum pH and temperatures for CuFeS_2_ were 3.0 and 60 °C, respectively. It can be seen that the CuFeS_2_ NPs have strong temperature tolerance. Then, the steady-state kinetics analysis was carried out using TMB and H_2_O_2_ as substrates. A typical Michaelis–Menten curve was obtained by varying the concentration of one substrate, TMB or H_2_O_2_, as the concentration of the other substrate was kept constant (Figure 3c,e). The linear equation (Figure 3d,f) was obtained by the double-reciprocal plotting method, Lineweaver–Burk, and the apparent kinetic parameters V_max_ and K_m_ were finally calculated and summarized in Appendix A. The results show that the behavior of CuFeS_2_ NPs on the substrates TMB and H_2_O_2_ followed the typical Michaelis–Menten equation. In addition, the CuFeS_2_ NPs have a lower K_m_ value and a higher V_max_ than natural horseradish peroxidase (HRP) and classical peroxidase analogs (Appendix A) [29,30], indicating that CuFeS_2_ NPs have a stronger affinity for substrates TMB and H_2_O_2_, as well as a higher catalytic efficiency. This may be due to the large specific surface area and small size of the nanoparticles, which can expose more surface-active sites.

### 3.4. Evaluation of Antibacterial Performance of CuFeS_2_ NPs

Because of the remarkable photothermal conversion ability and peroxidase activity of CuFeS_2_ NPs, MRSA, ESBL, and PA were selected as the study objects to evaluate the antibacterial performance of CuFeS_2_ NPs. In the absence of NIR irradiation, after treatment with CuFeS_2_ NPs +H_2_O_2_, the amount of bacterial colonies was expressively less than other groups (Figure 4). This was because of the conversion of low concentration H_2_O_2_ into highly toxic hydroxyl radical catalyzed by CuFeS_2_ NPs peroxidase, which increased the killing ability of bacterial cells. However, the antimicrobial ability of this strategy is limited, and many bacteria remained. In order to improve the antibacterial efficiency, NIR illumination can achieve PTT treatment and improve the catalytic efficiency of peroxidase. It can be seen from the antibacterial experiment results that, under the NIR irradiation, CuFeS_2_ NPs + H_2_O_2_ group significantly improved the antibacterial effect. It was calculated by the plate counting CuFeS_2_ NPs (2 μg mL^−1^) on 1 × 10^6^ CFU mL^−1^ MRSA, ESBL, and PA, with an antibacterial rate of more than 99%, which is superior to those of the other control groups and the conventional CuO NPs (4 mg mL^−1^) reported in the literature [31]. Thus, the antimicrobial strategy, based on the combination of PPT and CDT, can effectively inhibit bacteria, while evading the use of high concentration of H_2_O_2_ and CuFeS_2_ NPs and effectively avoiding the toxic and side effects. In order to evaluate the PPT/CDT synergistic antibacterial performance of CuFeS_2_ NPs and integrity of bacterial structure, live/dead staining assays were carried out (Appendix A). It was found that there was only a small amount of red fluorescence signals of the MRSA, ESBL, and PA cells after being treated with PBS, H_2_O_2_, CuFeS_2_, PBS + NIR, H_2_O_2_ + NIR, or CuFeS_2_ + NIR, indicating that the bacterial activity was normal. In contrast, the red fluorescence signals of the three bacterial cells were significantly enhanced after CuFeS_2_ + H_2_O_2_ treatment, indicating that the bacterial cell wall was seriously damaged, and the dye was bound to the nucleus. When NIR light was introduced, the red fluorescence signals of CuFeS_2_ + H_2_O_2_ + NIR-treated bacteria were enhanced, which proved that NIR light enhanced the antibacterial effects. These results were in line with the in vitro antibacterial experiments and SEM characterization, which all certified the enhanced antibacterial ability of CuFeS_2_ NPs under NIR light.

In order to reveal the antibacterial mechanism, DCFH-DA was used as an ROS fluorescent probe, which emitted green fluorescence after capturing ROS, for characterizing the difference of bacterial ROS in the experiment. The three strains treated by CuFeS_2_ + H_2_O_2_ showed a strong fluorescence signal, indicating that H_2_O_2_ decomposed into a hydroxyl radical with stronger oxidation under the catalysis of CuFeS_2_ peroxidase (Figure 5a,b). The ROS fluorescence signals in the three strains all increased significantly, indicating that the production of hydroxyl radical was enhanced with the assistance of NIR light, which led to an increase the ROS level in the bacteria, thus achieving better bacteriostatic effect. These results were in accordance with the in vitro bacteriostatic experiments. To further explore the mechanism of bacterial death, the cell morphology of MRSA, ESBL-E. coli, and PA were characterized by SEM. As can be seen from Figure 5c–e, the surface of bacterial cells treated with CuFeS_2_ + H_2_O_2_ was rough and wrinkled, or even ruptured. The cell structure of ESBL and PA was seriously damaged, and large areas of cavities could be clearly observed, while MRSA was characterized by severe bacterial shrinkage and a rough cell surface. In Appendix A, we added the SEM image of bacteria after the CuFeS_2_ + H_2_O_2_ + NIR and CuFeS_2_ + H_2_O_2_. The above results proved that both the Cu^2+^ release and ROS storm caused by CuFeS_2_ NPs peroxidase could cause severe damage to the cell structure, or even rupture, resulting in the death of bacteria. In addition, many particles were found to be attached to the surface of bacteria, which may be caused by the aggregation of the CuFeS_2_ NPs on the surface of cells. Meanwhile, the interaction between the CuFeS_2_ NPs and bacteria is conducive to the maximum killing of bacteria. In addition, the CuFeS_2_ NPs and GSH were co-incubated at 37 °C in phosphate buffer (pH 7.4, 0.1 M), and DTNB was used at the characteristic peak of GSH at 412 nm. The results showed that, with the increase of co-incubation time (0, 1, 3, 6 h), the UV absorption peak at 412 nm gradually weakened (Appendix A); when incubated for 6 h, the peak value was close to that of the PBS control group, which proved that CuFeS_2_ NPs could consume GSH under physiological conditions, and this GSH consumption ability might be attributed to the presence of Fe(III) in CuFeS_2_ NPs [32]. In the bacteriostatic process, the decrease in GSH level directly leads to the decrease of the bacteria’s antioxidant level, which is conducive to the production of ROS catalyzed by peroxidase, thus producing a stronger antibacterial ability.

### 3.5. CuFeS_2_ NPs for the Treatment of Wound Infection in Rats

Through a series of characterization, it has been proven that CuFeS_2_ NPs have excellent antibacterial ability. In order to evaluate the therapeutic ability of CuFeS_2_ NPs in the process of promoting wound healing, we established the bacteria-infected wound healing model with an MRSA-infected rat back wound. A total of eight groups (three rats per group) were set up. Under the condition with or without the irradiation of NIR, the wounds of rats were treated with PBS, H_2_O_2_, CuFeS_2_, and CuFeS_2_ + H_2_O_2_, respectively. Due to the PPT/CDT synergistic antibacterial performance and GSH consumption ability under the physiological conditions, CuFeS_2_ could produce more ROS to inhibit bacteria and promote wound healing. In Figure 6a, the wound healing of rats at different time points (1, 3, 5, and 7 days) was compared. As expected, the CuFeS_2_ + H_2_O_2_ + NIR (NIR 5 min, 1.0 W cm^−2^, 808 nm)-treated rats had the best wound healing in all the control and experimental groups. In addition, the wound area memory of rats after 7 days of treatment was statistically analyzed, and the treatment effect was quantified based on the wound size on day 1 (Figure 6b). After CuFeS_2_ + H_2_O_2_ + NIR treatment, the wound healed faster, and the wound area was the smallest. This proves that the PPT/CDT synergistic antibacterial system based on CuFeS_2_ has great application potential in practical antibacterial applications. At the same time, on the 7th day, the wound tissues of the rats with different treatments were taken, and the wound healing was analyzed via the H&E staining of tissue sections. In Figure 6c, no significant inflammation was found in the CuFeS_2_ + H_2_O_2_ + NIR-treated wound tissue, indicating the good antibacterial and anti-wound infection ability of the system. In addition, compared to the control group, the new epithelial tissues in the CuFeS_2_ + H_2_O_2_ + NIR group were thicker, and the inflammatory cells were significantly reduced, proving that the PPT/CDT synergistic antibacterial system of CuFeS_2_ effectively inhibited wound infection and promoted the growth of new tissues.

### 3.6. Biocompatibility of CuFeS_2_ NPs

In vitro and in vivo antibacterial experiments have certified that the CuFeS_2_ NPs-based PTT/CDT antibacterial system has excellent antibacterial properties and can make infected wounds heal more easily. Therefore, it is indispensable to value the biosafety of CuFeS_2_ NPs. We used rat red blood cells as experimental materials to study the hemolysis characteristics of CuFeS_2_ NPs; no significant hemolysis was discovered, even in the presence of high concentration of CuFeS_2_ NPs (100 μg mL^−1^), and the hemolysis rate of CuFeS_2_ NPs was calculated to be about 1.4%, indicating that CuFeS_2_ NPS had good blood compatibility (Appendix A). Appendix A results showed that the L929 cells survival rate kept at about 100%, even while the CuFeS_2_ NPS concentration reached 32 μg mL^−1^, which proved that CuFeS_2_ NPs had almost no toxicity to normal cells and good biocompatibility.

In addition, blood biochemical markers were measured by intravenous injection of CuFeS_2_ NPs into healthy rats (Figure 7a), and the results showed that blood indexes were normal, compared to the control group treated with PBS. As shown in Figure 7b, a histological analysis of the internal organs (kidney, spleen, liver, lung, and heart) of the rats showed no significant damage or inflammation. In conclusion, the prepared CuFeS_2_ NPs have good biocompatibility and are a potential antibacterial material.

## 4. Conclusions

In this work, CuFeS_2_ NPs with small and uniform particle sizes were prepared at room temperature. The CuFeS_2_ NPs have high peroxidase catalytic activity, excellent photothermal conversion ability, and good biocompatibility. Based on these excellent characteristics, the PTT/CDT synergistic antibacterial system of CuFeS_2_ NPs was constructed, and the experimental results showed that the antibacterial system has high-efficiency and broad-spectrum antibacterial ability. Under NIR illumination, it can decompose a low concentration of H_2_O_2_ into hydroxyl radicals under the catalysis of the internal peroxidase-like activity, thus achieving the purpose of a multimodal antibacterial, through the combination of near-distance ROS-mediated CDT and PTT. In addition, CuFeS_2_ NPs can consume the GSH in bacteria and destroy the antioxidant balance of cells, which is more conducive to the generation of oxidative damage. Finally, the bacteriostatic rate of CuFeS_2_ NPs (2.0 μg mL^−1^) and H_2_O_2_ (100 μM) was more than 99%, which assured the role of copper in the antibacterial effect, while reducing the agent amount significantly. In addition, in the treatment of MRSA wound infection in rats, the combined antibacterial system effectively inhibited the wound infection and promoted wound healing. More importantly, no obvious damage was observed to the internal organs of rats, which fully proved the safety of the antibacterial system. Therefore, this work proposes a safe and efficient antibacterial system, which brings new possibilities for fighting pathogenic bacteria and bacterial resistance and will greatly promote the further application of nanozymes in the treatment of bacterial infections and other diseases.

## Data Availability

The data presented in this study are available on request from the corresponding author.

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
