# Peer review of "Photothermal Regulated Nanozyme of CuFeS2 Nanoparticles for Efficiently Promoting Wound Healing Infected by Multidrug Resistant Bacteria"

_nanomaterials, 2022, doi:10.3390/nano12142469_

Round 1
Reviewer 1 Report
Nanomaterials 1789375
In this paper, the authors show the preparation of CuFeS2 nanoparticles (NPs) nanozyme antibacterial system with photothermal effect, and catalytic activity was proposed. The photothermal can regulate the CuFeS2 NPs close to their optimal reaction temperature (60 ºC) and in low concentration of H2O2 (100 μM). This study demonstrates that present excellent antibacterial ability and good biocompatibility of CuFeS2 NPs.
This paper is the sufficiently quality for to be published in Nanomaterials. The introduction is well documented. The work and paper is well organized, though the results are not spectacular. Only, is necessary that the authors complete the references.
Comment
1) Line 92, change presene for presence
2) Line 152, change 710.7, 161.9 for 710.70, 161.90. Revise all manuscript
3) Line 197, change 50 for 50.0
Author Response
1. Line 92, change presene for presence.
Response: Thanks for your considerate comments. We have revised the manuscript very carefully based on your valuable comments.
2. Line 152, change 710.7, 161.9 for 710.70, 161.90. Revise all manuscript.
Response: Thanks for your comments. We have revised the manuscript.
3. Line 197, change 50 for 50.0
Response: Thanks for your valuable comment. We have revised the manuscript very carefully based on your comments.
Reviewer 2 Report
The authors have carried out their research on the preparation and estimation of CuFeS2 Nanoparticles for Efficiently Promoting Wound Healing Infected by Multi-3 Drug-Resistant Bacteria.
There are the following comments as follows:
1. In the L-12, it is recommended to be included the background link of CuFeS2 and wound healing.
2. In the introduction section, L-36, it is recommended to include these references which are related to microorganisms and nanocarriers. such as Chitosan-gum arabic embedded alizarin nanocarriers inhibit biofilm formation of multispecies microorganisms. Carbohyd. Polym.. 2022 May 15;284:118959.
3. For the synthesis of CuFeS2 whether authors optimized the method?
4. The image a and b in Fig1. was not cleared. it is recommended to include high-resolution images with 10nm and 100nm scales.
5. In Fig 5 c, d, and e, it is recommended to be included 10µm 50 µm images and explain cell disruption.
Reviewer 3 Report
Row 37-41 - It is not clear whether the low toxicity is an advantage, how it is determined - against bacteria (what?), or in a living organism;
Rows 42-43 - Where are these nanosomes applied - in animals, humans, cells?
Rows 45-46 - Healthy tissues in animals or humans?
Rows 51-55 - what kind of photothermal therapy are we talking about? In humans, in animal models, in what exactly are tumors?
- Is damage to healthy tissue reversible after NIR exposure?
The introduction is confusing - it is not clear what model the authors are targeting - the cellular system, animal, human?
At the end of the Chapter Introduction, there is no precise and clearly formulated goal - this does not allow to assess at the end of the article whether the goal is met and what conclusions can be drawn?
In the chapter "Materials and methods" - it is not clear what exactly the object is - in the title, it says “mice” - 2.4. The mouse would model, and below - Sprague-Dawley (SD) rats - which is true?
The chapter "Materials and methods" does not describe the pattern of erythrocyte hemolysis - what exactly is the hemolysis reported - osmotic (depending on the concentration) or acidic (pH)?
What is meant by Blood biochemical index? It is not described in the Materials and Methods, nor in the text below the figures. Why was this indicator chosen and what information does it carry?
Reviewer 4 Report
The manuscript authored by Zezhong Liu reports the fabrication of CuFeS2 nanoparticles, their physico-chemical characterization and application as antibacterial and wound healing agents. Although the original idea is interesting, there are some major revisions that need to be addressed.
1) English should be strongly revised. There are several grammatical errors along the whole manuscript, as well as typing errors. Abbreviations have to be defined the first time they are mentioned, and units need to be unified choosing only one format, mg/mL or mg mL-1, along the whole manuscript.
Apart from that, there is a duplicated sentence in lines 383 and 384.
I also suggest to include all the materials and methods description in the manuscript, instead of incorporating a part in the main manuscript and another part in the supporting information to make it more readable.
2) Importantly, references section has to be double checked because there are several references with the same number. Moreover, the journal names should appear in the abbreviated form.
3) Section 2.1 should be entitled as Synthesis of CuFeS2 nanoparticles instead of CuFeS2 nanozymes as mentioned in the rest of the manuscript.
4) Concerning NPs concentrations shown in Fig. 1h, is the heating determined by the dispersion of nanoparticles in PBS? This needs to be clarified.
5 5) Graphs in Fig. 2e and 2f should show the same x-axis scale from 400 to 1000 nm to prove that the absorbance increase takes place in the whole range.
6) I suggest to increase the brightness of the image corresponding to CuFeS2 NPs+H2O2 in order to better show the p. aureginosa bacteria that are present in the petri dish, and add the scale bar in the pictures.
7) Pictures for p. aureginosa in Fig. 5a cannot be correlated with those spectra in Fig. 5b because no fluorescence is observed in the pictures. In addition to this, SEM images of bacteria after NIR treatment should be included to compare the bacteria state without and with NIR treatment.
n 8) I have a serious concern about the heating of the CuFeS2 nanoparticles after NIR treatment, which achieves values higher than 42ºC. Temperatures higher than 42ºC produce irreversible damages in the healthy cells, which would be a serious drawback for the employment of these materials (CuFeS2 nanoparticles) in presence of NIR light. In this sense, L929 cell viability tests in presence of NIR light needs to be performed and incorporated in the manuscript for comparison with those ones without NIR shown in Fig. S7.
Round 2
Reviewer 2 Report
The revised manuscript is suitable for further process in the journal.
Author Response
Thanks for your considerate comments. We have revised the manuscript very carefully based on your valuable comments.
Reviewer 3 Report
The authors have complied with the reviewer's opinion to a large extent.
However, again, the purpose of the present work is not apparent - yes, the research is necessary, and there is a lack of information on this issue in the scientific literature, but it is not clear that you, in this very article, try to solve this problem.
The use of the concept of the index has a slightly different meaning - coefficient, dependence between values, the term "biochemical markers" is more appropriate. Phosphatases and transaminases are markers, not indices. Please refine the expression.
Author Response
Thanks for your good comments and kind recommendation. The notable increase in bacterial resistance to current available antibiotics has led to a need to develop alternative antibacterial agents. Nanoparticles (NPs) have unique physicochemical properties and have emerged as alternative tools to control bacterial infection and overcome antibiotic resistance. We use CuFeS2 nanozyme to kill common bacteria in life and prevent wound infection in animals. We have corrected the problem you pointed out.
Reviewer 4 Report
The manuscript can be accepted for publication in its current form.
Author Response

(The authors gave the same response as above.)
